# Access to primary care for children and young people (CYP) in the UK: a scoping review of CYP's, caregivers' and healthcare professionals' views and experiences of facilitators and barriers

Lauren Herlitz ,[1] Emily Ashford,[2] Claire Powell ,[1] Kevin Herbert ,[3] Stephen Morris ,[3] Jenny Woodman [2]

¹NIHR Children and Families Policy Research Unit, Population, Policy and Practice, UCL GOS Institute of Child Health, London, UK
²Thomas Coram Research Unit, UCL Social Research Institute, London, UK
³Public Health & Primary Care, University of Cambridge, Cambridge, UK

**Correspondence to**
Dr Lauren Herlitz;
l.herlitz@ucl.ac.uk

## ABSTRACT

**Objectives** To examine children and young people's (CYP), caregivers' and healthcare professionals' (HCPs) views or experiences of facilitators and barriers to CYP access to UK primary care services to better understand healthcare inequity. To explore differences across CYP subpopulations with greater health needs from deprived areas, identifying as ethnic minorities, with experiences of state care, special educational needs or disabilities, chronic conditions or mental health problems.

**Design** Scoping review.

**Eligibility criteria** Included studies were in English, published 2012–2022 and reported: the views/experiences of CYP (0–25 years), caregivers or HCPs about accessing UK primary care; using quantitative or qualitative empirical methods.

**Data sources** PubMed, CINAHL, Web of Science, PsycINFO and Scopus.

**Results** We included 47 reports (46 studies). CYP/caregivers' decision to access care was facilitated by CYP/caregivers' or their family/friends' ability to identify a health issue as warranting healthcare attention. Barriers to accessing care included perceived stigma (eg, being seen as a bad parent), embarrassment and discrimination experiences. CYP and caregivers believed longer opening hours could facilitate more timely access to care. Caregivers and HCPs reported that delayed or rejected referrals to secondary or adult care were a barrier to having needs met, especially for CYP with poor mental health. CYP and caregivers in numerous studies emphasised the importance of communication and trust with HCPs, including taking their concerns seriously, being knowledgeable and providing continuity of care for CYP. Common barriers reported across high-need subpopulations were caregivers needing knowledge and confidence to advocate for their child, gaps in HCP's knowledge and a lack of connectedness between primary and secondary care.

**Conclusions** Connecting general practices and community health workers/services, improving CYP/caregivers' understanding of common childhood conditions, addressing HCP's knowledge gaps in paediatric care and integrated approaches between primary and secondary care may reduce inequity in access.

## STRENGTHS AND LIMITATIONS OF THIS STUDY

⇒ The review was rigorously conducted and included quality appraisal.
⇒ Mapping patterns of facilitators/barriers across different subpopulations with higher health needs was a strength of the review, revealing that access was affected by caregivers having to be able to confidently advocate for their child's needs, multiple barriers existed for some groups and there was a lack of evidence on access to looked-after children.
⇒ Studies in systematic reviews were not screened and we did not search for grey literature due to time and resources constraints.
⇒ Workforce-related barriers, for example, recruiting and retaining general practitioners, which affect both children and young people's and adult patients were not identified using our search terms.

## INTRODUCTION

Access to healthcare can be defined as the opportunity to identify healthcare needs, to seek, reach and use healthcare services, and to have healthcare needs met.[1 2] Primary care access in childhood is important to ensure that children and young people (CYP) are vaccinated, reach developmental milestones, are safeguarded, and that acute and chronic conditions are identified and managed.[3 4] Evidence also suggests that improved access to primary care may reduce the escalation of health concerns, alleviating pressure on secondary care.[5–7] The National Health Service's (NHS) Long Term Plan in England highlights the role of primary care in reducing health inequalities and ensuring CYP (aged 0–25) have a strong start in life, in particular improving access for CYP with mental health problems, learning disabilities or autism.[8] Unmet healthcare needs in

BMJ

adolescence are an independent predictor of poor adult health.[9 10]

Recent evidence suggests that CYP access to primary care is inequitable. For example, UK cohort studies linked to routine health data found that CYP living in deprived areas were less likely to access primary care relative to their wealthier peers and more likely to use acute care.[11–13] Inequalities in CYP access to care may result from variation in the supply of healthcare by area deprivation[14]; differences in how conditions are identified and managed, for example, because of increased multimorbidity in CYP in deprived areas[15] or variation in healthcare professionals' (HCPs) expertise.[16] Marginalised CYP and caregivers may not identify themselves as requiring health treatment or may lack knowledge of available healthcare services and how to navigate complex healthcare systems.[17 18] CYP's access is also affected by age and development, with younger children reliant on caregivers, and older adolescents and young adults seeking services independently.[19]

Systematic reviews have been conducted on CYP and HCP's views of some specific healthcare services in the UK.[20–22] In 2021, the National Institute for Health and Care Excellence (NICE) published guidelines on babies, children and young people's experience of healthcare, which included an evidence review of healthcare access (including acute, primary and secondary care settings).[19] Focusing on CYP under 18, it found that a key barrier was a lack of information about when to access healthcare services, what services were available and how CYP could be supported to access them. CYP also reported that they could avoid seeking help due to fear of being blamed, labelled or being embarrassed, or because they were unsure about the limits of confidentiality.[19] Building on evidence from the NICE review, this study focused specifically on CYP's access to primary care, synthesising perspectives of CYP, caregivers and HCP across primary care services in order to deepen understanding of healthcare inequity, barriers to healthcare and how to address them, and looked in detail at facilitators and barriers for CYP with high health needs.

## METHOD

Our methods were informed by rapid evidence review guidance.[23] We preregistered the review protocol in the Open Science Framework (https://osf.io/mfc3z). The study followed the Preferred Reporting Items for Systematic Reviews and Meta-Analyses extension for Scoping Reviews statement (online supplemental additional file 1).

### Inclusion/exclusion criteria

We included a study if it:

► Reported the views or experiences of CYP (aged 0–25 years), caregiver (ie, parent or carer) or HCPs on the facilitators and barriers to primary care access,

including studies that examined primary care as a means of accessing secondary care.
► Was based in the UK.
► Used quantitative or qualitative empirical methods.
► Was published in English between 2012 and 2022.

We excluded studies that focused on access to school health services, access to primary care during the COVID-19 pandemic or on the uptake of vaccinations/immunisations. We excluded systematic reviews.

### Search strategy

We searched PubMed, CINAHL, Web of Science (Social Sciences Citation Index), PsycINFO and Scopus using free-text and index terms for the following concepts: healthcare access, primary care, CYP, UK and facilitator and barriers (see online supplemental additional file 2).

### Document selection

We imported the search results into Rayyan software (https://www.rayyan.ai/) for deduplication and screening. Five reviewers independently conducted title/abstract screening and twenty per cent (N=1334) were checked by a second reviewer. Two reviewers independently conducted full-text screening and 25% (N=36) were checked by a second reviewer. The first and second reviewers discussed disagreements until a consensus was reached, bringing in a third team member where necessary.

### Data extraction

The following data were extracted: study sample/population; primary care setting; area of healthcare; study design/methodology; factors affecting primary care access. Data on access to primary care during the COVID-19 pandemic were not extracted.

### Quality appraisal

Five reviewers assessed study quality using the Mixed Methods Appraisal Tool.[24 25] No study was excluded based on quality, but study quality is acknowledged in the findings and quotes presented are from medium-quality and high-quality studies only. One reviewer assigned studies two 'weight-of-evidence' ratings,[26] one for quality and one for relevance to answering the review question, rated 'low', 'medium' or 'high' (see online supplemental additional file 3). For a judgement of 'high' relevance, studies had to describe, with breadth and depth, factors influencing primary care access and privilege participants' perspectives.

### Data synthesis

Data were synthesised using framework analysis[27] to systematically review and map the data from each study using a structured template (see online supplemental additional file 4). After data were descriptively coded, a conceptual framework was applied following a patient pathway from a CYP/caregiver identifying a health issue and deciding to seek help, to organising an appointment and attending a consultation, influenced by previous

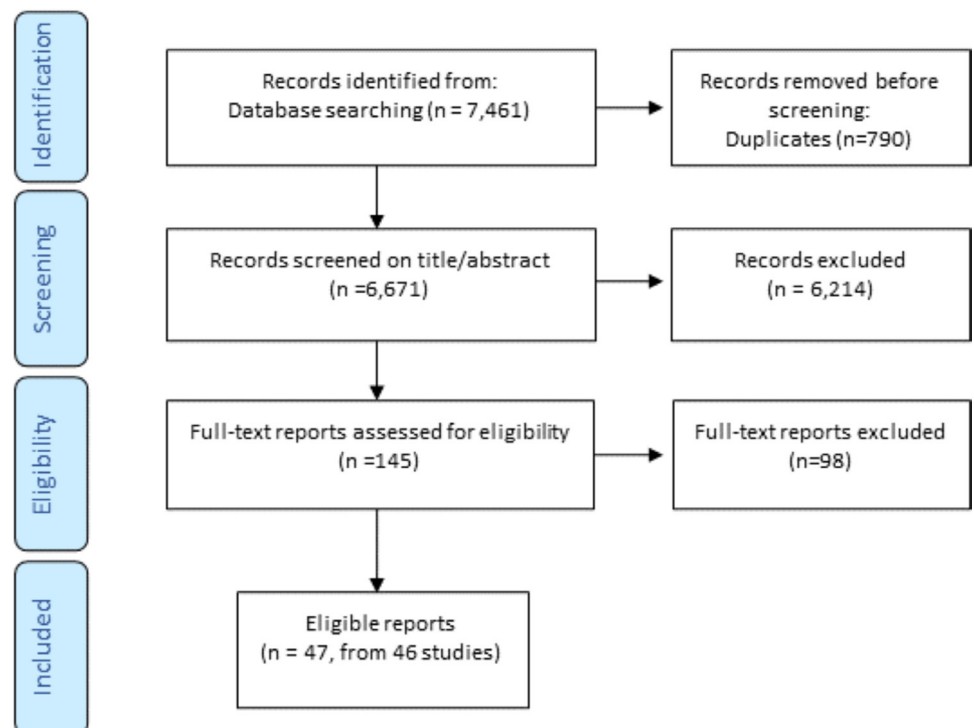

**Figure 1** PRISMA flow diagram. PRISMA, Preferred Reporting Items for Systematic Reviews and Meta-Analyses.

work.[28] To visualise whether any codes and themes were pertinent for specific subpopulations with high needs, data were colour-coded for the following CYP groups from deprived areas, experiences of state care (ie, looked-after children), identifying as ethnic minorities, with SEN or disabilities, with chronic conditions and with mental health problems. Subpopulations were selected from CYP target populations and focus clinical areas in the 'Core20Plus5', the NHS England strategy for reducing health inequalities.[29] Subthemes reported for these subpopulations were systematically mapped.

### Patient and public involvement
Patients and/or the public were not involved in the design, conduct or reporting of this review.

### RESULTS
Of the 6671 unique titles/abstracts generated from database searching in February 2022, 47 reports (of 46 studies) met the inclusion criteria (see figure 1).

### Study characteristics
#### Study design/methods
Most studies were qualitative using interviews (n=25), or focus groups (n=6), or focus groups and interviews (n=5). All quantitative studies used cross-sectional surveys (n=5) while mixed-method studies used surveys that contained open and closed questions (n=5) (see table 1).

#### CYP age focus and health topic
10 studies (22%) focused on CYP under 5 years, 12 (26%) were about CYP between the ages of 0 and 15 years, 10 (22%) focused on young people (YP) aged 16–25 years and the rest focused on a range of different ages between 0 and 25 years (see online supplemental table).

13 studies (28%) were related to CYP access for non-specific health conditions; 11 (24%) were about CYP with mental health conditions; 8 (17%) were about CYP's oral health; 4 (9%) focused on CYP with chronic health conditions; 4 (9%) were about CYP with physical health conditions; 4 (9%) focused on YP's sexual health; 1 (2%) was on help-seeking for children's gender identity and 1 (2%) examined CYP eye care from optometry practices (see table 1).

#### Study participants
Most studies invited either caregivers (n=18), YP (aged 11+ years) (n=11) or HCPs (n=10) to participate; seven studies included more than one type of participant and one study surveyed optometry practices. More than half of the studies focused on CYP in general (n=28); the rest focused on a particular subpopulation(s) (see online supplemental table).

#### Primary healthcare setting
The following healthcare settings were studied (note, several studies covered multiple settings): general practice (n=27), health visiting (n=8), dental care (n=6), overall primary care (excluding dental care or optometry) (n=4), pharmacy services (n=3), optometry (n=1), walk-in centres (n=1) or sexual health clinics (n=1) (see table 1).

**Table 1** Characteristics of studies included

| Author (year) (citation) location | Primary healthcare setting: main focus of study | Design |
|---|---|---|
| Ahmaro et al (2021) England[52] | Pharmacies: sexual health and chlamydia testing and chlamydia treatment. | Qual; I |
| Alexakis et al (2015) England[39] | General practice: needs of YP with inflammatory bowel disease from black and ethnic minority communities. | Qual; I |
| Appleton et al (2022) England[68] | General practice: receiving primary care support after child and adolescent mental health services. | Qual; I |
| Bosley et al (2021) England[58] | General practice and health visiting: the accessibility and expertise of HCPs. | Qual; FG and I |
| Brigham et al (2012) England[95] | Health visiting: health visitors' (HVs) perceptions of their role and skills, sharing expertise and work with other agencies. | Qual; FG |
| Coleman-Fountain et al (2020) n/k[34] | General practice: exploring how autistic young adults understand and manage mental health problems. | Qual; I |
| Condon et al (2020) England[45] | General practice and health visiting: using services postmigration from Romania, Poland, Pakistan or Somalia. | Qual; FG |
| Corry and Leavey (2017) N. Ireland[51] | General practice: adolescents' attitudes to consulting their GP about psychological problems. | Qual; FG |
| Coyle et al, (2013) N. Ireland and Scotland[96] | Dental care: HCP's willingness to treat adolescents with learning disabilities (LD) in primary dental care. | Quant; S |
| Crocker et al, (2013) Wales[56] | General practice: consulting a GP before the day of hospital presentation with pneumonia or empyema. | Mixed; S and I |
| Crouch et al (2019) England[65] | General practice: seeking help and accessing specialist treatment for childhood anxiety | Qual; I |
| Dando et al,(2019) England[46] | General practice: healthcare experiences of Albanian survivors of modern slavery and sexual exploitation | Qual; I |
| Davey et al (2013) England[60] | General practice and walk-in centres: the needs and experiences of young adults of primary healthcare services. | Qual; I |
| Dickson (2015) N. Ireland[61] | Dental care: parents' perceptions of factors influencing dental registrations of children living within a Sure Start area. | Qual; I |
| Diwakar et al (2019) England[66] | General practice: understanding parent experiences with paediatric allergy pathways. | Qual; I |
| Eskytė et al (2021) England[36] | Health visiting: organisational factors that obstruct HVs from speaking to parents of babies about oral health | Qual; I and FG |
| Fox et al (2017) England[40] | General practice and health visiting: health, education and social care services support for CYP with autism. | Qual; I |
| Fox et al (2015) England[69] | General practice: identifying barriers to and enablers for discussing self-harm with YP. | Mixed; online S and I |
| French et al (2020) UK[42] | General practice: exploring the primary care experiences of referral and management of ADHD | Qual; I |
| Henderson and Rubin (2014) England[37] | Dental care: an oral health promotion initiative to improve access for pre-school children in deprived communities. | Qual; FG and I |
| Ingram et al (2013) n/k[30] | General practice: support/information needs when children have respiratory tract infections with a cough | Qual; FG and I |
| Jobanputra and Singh (2020) England[70] | General practice: exploring GPs' views on the management of adolescents with mental health disorders | Qual; I |
| Jones et al (2017) England[53] | General practice: receiving chlamydia testing with condoms, contraceptive information and HIV testing. | Qual; I |
| Lewney et al (2019) England[67] | Health visiting: HVs views about providing oral health advice and dealing with dental issues | Qual; I |
| McDonagh et al (2020) UK[54] | General practice: barriers to chlamydia testing and potential intervention functions and implementation strategies. | Qual; I |
| Mughal et al (2021) England[35] | General practice: help-seeking behaviours, GP care and healthcare access for YP who self-harm. | Qual; I |

Continued

**Table 1** Continued

| Author (year) (citation) location | Primary healthcare setting: main focus of study | Design |
|---|---|---|
| Muirhead et al (2017) England[62] | Dental care: foster carers' oral health knowledge, attitudes and experiences of managing foster children's oral health. | Qual; FG |
| Neill et al (2016)* England[31] | Primary care (all except dental and optometry): making decisions during acute childhood illness at home. | Qual; FG and I |
| Neill et al (2015)* England[32] | Primary care (all except dental and optometry): information resources for decision-making in acute childhood illness at home. | Qual; FG and I |
| O'Brien et al (2019) England[71] | General practice: identifying, managing and accessing specialist services for anxiety disorders. | Quant; S |
| O'Brien et al (2017) England[49] | General practice: identification, management and access to specialist services for anxiety disorders. | Qual; I |
| Ochieng (2020) England[47] | Health visiting: factors that either influence healthy weight in black African children. | Qual; FG |
| Rapley et al (2021) England[59] | Primary care (all except optometry): experiences of care, from initial symptoms to initial referral to paediatric rheumatology. | Qual; I |
| Rashed et al (2022) England[57] | Pharmacy and general practice: exploring the experiences, barriers and recommendations of caregivers and YP regarding the use of community pharmacies for children. | Mixed; S |
| Redsell et al (2013) England[73] | Health visiting: the beliefs and practices of UK HVs concerning infants at risk of developing obesity. | Qual; I |
| Rickett et al (2021) Scotland, Wales and England[43] | General practice: healthcare expectations and experiences of caregivers seeking support for their gender-diverse children | Mixed; S |
| Roberts et al (2014) England[74] | General practice: GPs' experiences and views of consulting with adolescents with psychological difficulties. | Qual; I |
| Roberts and Condon (2014) England[38] | Dental care: exploring parental attitudes to pre-school oral health. | Qual; I |
| Salaheddin and Mason (2016) UK[50] | General practice: exploring the barriers to accessing mental health support among young adults. | Mixed; S |
| Satherley et al (2021) England[41] | General practice: how mothers living in deprived neighbourhoods support their children with health conditions. | Qual; I |
| Turnbull et al (2021) England[55] | Pharmacy and sexual health clinic: accessing emergency contraception pills | Qual; I |
| Turner et al (2012) England[44] | General practice: views and experiences of primary care as a treatment setting for childhood obesity. | Qual; I |
| Usher-Smith et al (2015) England[33] | General practice and secondary care: Explored the pathway to diagnosis of type one diabetes. | Quant; S |
| Williams et al (2014) England and Wales[63] | Dental care: the impact of a community-based dental care pathway on children's dental care entering residential or foster care. | Qual; I and RDC |
| Williams et al (2012) England[48] | Not specified (preventative primary care services): African and African-Caribbean fathers' beliefs about fatherhood, health and preventive primary care services. | Qual; FG |
| Wilson et al (2021) England[64] | Optometric practices: accessibility of eye care for children with typical development and those with autism. | Quant; S |
| Yassaee et al (2017) England[72] | General practice: GP experiences, associations between poor reported GP experience and physical and mental health measures and service utilisation. | Quant; S |

*Reports are from the same study.
ADHD, attention deficit hyperactivity disorder; CYP, children and young people's; FG, focus groups; GP, general practitioner; Qual, qualitative; Quant, quantitative; RDC, routine data collection; YP, young people.

## Study quality and relevance

10 studies (22%) were rated high on both quality and relevance (see online supplemental additional file 3). Studies on CYP with chronic conditions and sexual health were rated higher on quality and relevance while half of oral health studies, and the only optometry study, were rated low on quality.

## Facilitators and barriers in CYP access to primary care

We constructed three overarching themes on CYP and caregivers' access to primary care: deciding to access care; reaching and entering services; and communication and trust between HCPs, caregivers and CYP (see figure 2). Online supplemental additional file 5 provides a table of themes by study.

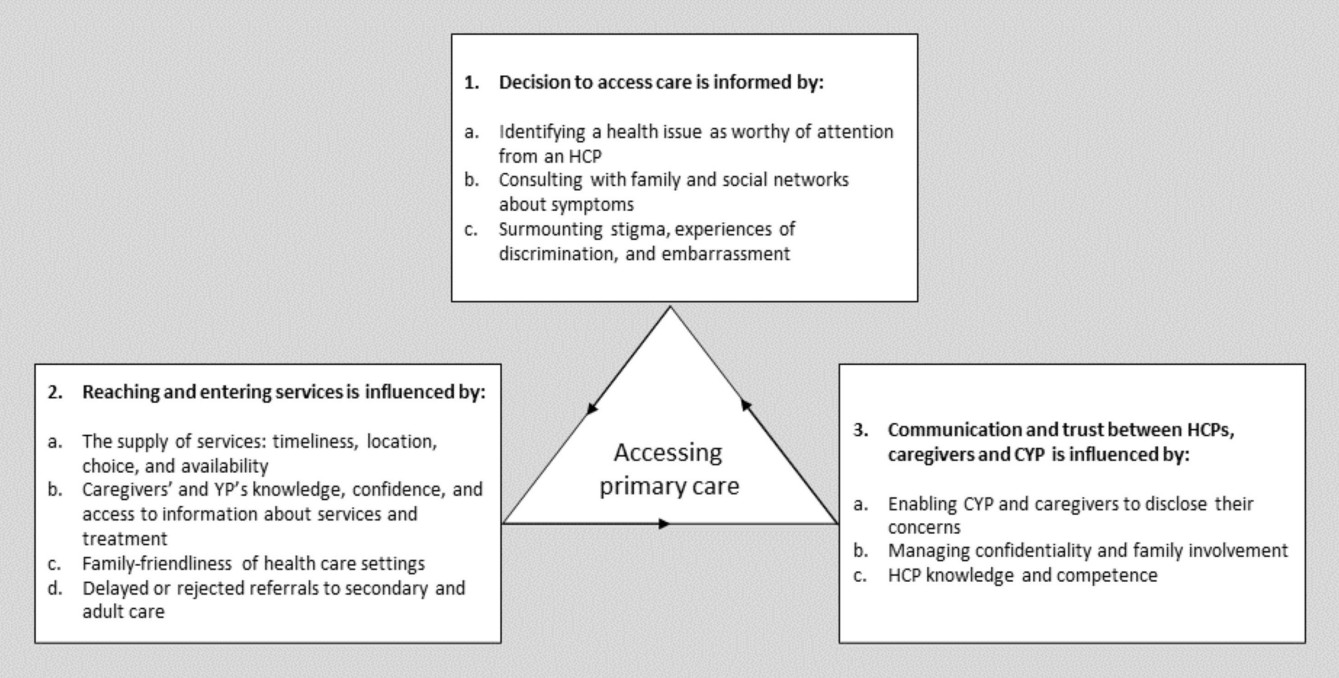

**Figure 2** Facilitators and barriers to CYP access to primary care. CYP, children and young people's; HCP, healthcare professional; YP, young people.

### Deciding to access care

Multiple studies examined caregivers' and YPs' decisions to access healthcare. We constructed three subthemes: identifying a health issue as worthy of attention from an HCP (n=9); consulting with family and social networks about symptoms (n=5) and surmounting stigma, experiences of discrimination and embarrassment (n=17).

#### Identifying a health issue as worthy of attention from an HCP

Three studies (of medium/high-quality, in four reports) reported that a YP or caregiver would only seek help if they considered their symptoms serious, wanting to avoid burdening health system resources.[30–33] As well as assessing severity, caregivers considered the familiarity of the illness, their child's level of distress, and whether symptoms were worsening and/or persisting.[30–33] First-time parents were more likely to access care as childhood illnesses were unfamiliar, making it difficult for parents to judge severity.[31] The basis of YP's (aged 16+) decision-making for mental health concerns was similar to those of caregivers; they would consider seeking help if their distress was severe and enduring, and was felt to be beyond self-management.[34 35]

In three studies of preschool children, (one high quality, two low quality), two of which focused on CYP from deprived areas, HVs and parents reported oral health was of low priority in comparison to assessing children's physical health and developmental milestones.[36–38]

#### Consulting with family and social networks about symptoms

Five studies (of medium/high quality, in six reports) reported that parents and YP utilised their family and social networks, as well as material resources (eg, websites, leaflets) to confirm their decision to consult.[30–32 39–41] Contradictory advice or encouragement from family/friends to seek help contributed to a decision to consult.[30 41]

In three high-quality studies (in four reports), caregivers from South Asian, Gypsy/Travelling, and Somali communities, and YP from Black and ethnic minority groups reported that they would defer to children's grandparents, extended family or community members for advice, and relied on their children or local community to relay information if they could not read and write in English.[31 32 39 40] Two of the studies identified that if the community were unfamiliar with the syndrome/illness studied (or it was stigmatised), families could encounter advice not to seek help, dismissive responses to diagnosis, or inappropriate efforts to treat the condition[39 40]:

> Some of the people say, "Why are you saying something silly like this?" He's a child, he will grow out of it [autism]. A lot of children can't talk at the normal age, why don't you wait? Don't go to the doctors. He will grow out of these things. (Caregiver,[40]).

#### Surmounting stigma, experiences of discrimination and embarrassment

Stigma, discrimination and embarrassment were reported as barriers to help-seeking. Four studies (of mixed quality) highlighted that parents could feel judged for their parenting, labelled as 'pushy parents' or blamed for their child's condition. This was found in studies of mothers of low socioeconomic status, children with attention deficit

hyperactivity disorder (ADHD), gender-diverse children and those experiencing childhood obesity.[41–44]

> I'm on income support, so asking me to feed her quinoas, avocados and vegetables, that's just not … I can barely get the milk for the tea. And then I have five other children, how am I going to measure the powder every meal? (Caregiver,[41])

Stigma and discrimination experienced by ethnic minorities and migrants were barriers identified in four studies (of mixed quality).[45–48] For example, caregivers being sent away or ignored[46] or labelled as 'aggressive' when trying to resolve misunderstandings with HCPs.[48] Two studies (of medium/high quality) reported that African/African-Caribbean fathers and migrant caregivers perceived preventative services as part of a government surveillance system, indicating distrust of services.[45 48]

Stigma related to mental health felt by YP or their caregiver could be a barrier to seeking help from a general practitioner (GP), as reported in four studies (of mixed quality),[35 40 49 50] two of which suggested that mental health stigma was more common among ethnic minorities.[40 49] Believing that they would not be taken seriously, or fears that they would not have a say in their treatment, were barriers to seeking mental health support reported by YP (aged 13+).[35 50 51]

Embarrassment was a common barrier for YP (aged 16+) seeking sexual healthcare, noted in four studies (of medium/high quality).[52–55] YP reported being concerned about being seen by family/friends or judged by staff, feeling ashamed to be accessing emergency contraception, and embarrassed by the testing procedure itself. In one high-quality study, YP felt HCPs might make assumptions about promiscuity or judge them on the basis of their sexuality, affecting their willingness to be tested in general practice.[54]

### Reaching and entering services

After caregivers or YP decided healthcare support was needed, organising an appointment and entering services was the next step to access. We identified four subthemes among the many studies exploring this theme: the supply of services (n=22); caregivers' and YP's knowledge, confidence and access to information about services and treatment (n=18); family-friendly healthcare settings (n=6) and delayed or rejected referrals to secondary or adult care (n=7).

### *The supply of services: timeliness, location, choice and availability*

Caregivers and YP reported that longer GP, pharmacy and sexual health clinic opening hours could facilitate more timely access to care in seven studies (of mixed quality).[32 52 54–57] Caregivers of young children (under 5) noted it could be difficult to attend (or phone for) appointments early in the morning when children were getting ready for school, or at children's bedtime, in one high-quality study.[31] Caregivers were willing to seek advice and treatment from nurses, pharmacists and NHS Direct

(instead of a GP) if they wanted to be seen quickly, and/or the illness was considered common and/or mild.[30 32 57 58] Two studies (of medium/high quality) found that parents sought out a private diagnosis to gain more timely access to care (for ADHD and for juvenile idiopathic arthritis) and to evade GP 'gatekeeping'.[42 59]

Healthcare practices that were within walking distance of patients' homes or work, or on bus routes could facilitate access, as reported by caregivers and YP,[52 53 56 58] as could colocating health and other children's services, according to HVs and caregivers.[36 58] Choice of healthcare settings and professionals was salient in sexual health studies; YP (aged 16+) appreciated options for seeking testing and advice (online, pharmacy, GP, sexual health clinic) where privacy/discretion was a key consideration, and some YP preferred to speak to a staff member with the same gender identity.[52–55 60]

In terms of service availability, participants from multiple studies reported long waiting times to see a GP.[33 35 53 56 57 60] Reduced engagement with HVs as a result of cuts to provision was noticed by caregivers and HVs in two studies (of medium/high quality).[36 58] HVs also noted the lack of NHS dentists in the deprived areas in which they worked.[36] Three studies (of low/medium quality) found caregivers had received conflicting information from dental practices about the age for registering children.[38 61 62] One low-quality study noted that strict non-attendance and deregistration policies to manage resources in dentistry adversely affected looked-after children, who often had a history of low dental attendance, poor diet and oral hygiene before care entry and higher dental care anxiety[63]:

> They haven't been to the dentist for a long time… then they are suddenly faced with a dental appointment, and often they are fine, and then the day before or the day of the appointment, they categorically refuse to go. (Caregiver,[63]).

One study (of low quality) found optometry practices varied in whether they thought young children (under 5) should be examined by a GP or an optometrist.[64]

### *Caregivers' and YP's knowledge, confidence and access to information about services and treatment*

Studies (of mixed quality) reported variation in caregivers' and YP's knowledge of appointment systems, though YP were more often inexperienced in accessing care.[35 37 40 45 58 60] In a high-quality study on Somali migrants' access to care for CYP with autism, caregivers reported feeling overwhelmed by the complexity of the health and education system, and the lack of clarity around the purpose of appointments and professional roles.[40] Caregivers of CYP with complex needs and HCPs reported that parents having the confidence to persist in asking for support for their child helped them to gain timely access to care and appropriate referrals to secondary care, as noted in multiple studies (of medium/high quality)

relating to CYP with chronic conditions, mental health problems, ADHD and gender diversity.[42 43 49 59 65 66]

> …if I felt a child was, not necessarily needing secondary care but the family were overly concerned and were pushing for a referral [for anxiety], I would probably [go] along with that. (GP,[49])

A lack of clear, visible information about what services were offered at the GP and pharmacy was reported by YP (aged 16+) and caregivers in four studies (of a mix of quality).[53 54 57 60] Two high-quality studies identified that confusion over who was responsible for organising an interpreter was a barrier to dental and GP care.[40 67] Some caregivers of young children reported that they liked to receive practical resources and hard copies of information about child health that they could refer back to, reported in two high-quality studies focused on CYP from deprived areas.[32 36] YP (aged 16+) reported they would like demonstration videos via websites alongside instructions for self-testing in one medium-quality sexual health study.[54]

### Family-friendly healthcare settings

The healthcare setting itself could be a barrier to help-seeking. It was stressful for caregivers of young children to wait with their child or with other children in tow, a problem particularly affecting single parents and parents without easy access to childcare.[31] In some practices, the physical environment could be difficult to navigate with a buggy.[38] Signalling that healthcare settings were child-friendly and parent-friendly, for example, by putting posters or toys in the waiting area for younger children,[38 57] or being warm and approachable at the reception desk, was appreciated by caregivers and YP, particularly caregivers who were not fluent in English or YP who were struggling with their mental health.[31 68 69] One medium-quality study flagged that the fathers in their study perceived child health services as designed for women, rather than men.[48]

### Delayed or rejected referrals to secondary or adult care

Delayed or rejected referrals to secondary or adult care were a barrier to CYP having their health needs met. Three studies (of medium/high-quality) about care for anxiety, ADHD, and juvenile idiopathic arthritis reported several reasons for GPs delaying referrals: a decision to 'wait and see' to see if more evidence materialised, the assumption that symptoms were the result of another non-medical cause or were due to a pre-existing known condition.[42 59 65] The feeling of being 'passed around' services was recounted by both HCP and caregivers of CYP with these conditions.[42 49 59]

Both caregivers' and HCPs described frustration over the care of CYP's mental health and ADHD resulting from long waiting lists for Child and Adolescent Mental Health Services (CAMHS); rejected referrals to CAMHS due to high thresholds, GPs lack of knowledge about available mental health and ADHD services, or what information is needed to obtain a successful referral; or lack of clear care pathways, reported in five studies (of mostly medium/high-quality).[42 49 69–71]

### Communication and trust between HCPs, caregivers and CYP in consultations

Once a consultation with an HCP professional was arranged, accessing the help CYP needed depended on communication and trust with HCPs. We constructed three subthemes from multiple studies: enabling CYP and caregivers to disclose their concerns (n=22); managing confidentiality and parental involvement (n=6); and HCP knowledge and competence (n=20).

### Enabling CYP and caregivers to disclose their concerns

A 2014 national survey of adolescents in England found that only 54% of YP who had visited the GP in the last year felt able to talk to them about personal matters.[72] Numerous studies highlighted that the quality of patient encounters with HCPs impacted on their willingness to disclose information. Caregivers and YP across many studies identified the same HCP attributes that would help them to share their concerns: HCPs should be reassuring, trustworthy and knowledgeable.[35 51 52 54 55 57 58 63 65]

> His [the GP's] patience and lack of judgement was amazing, just to listen to my experiences of what happens for emotionally when I'm self-harming… it was incredible. (YP, 22 years,[35])

HCPs showing that they were listening and taking CYP's symptoms seriously was very important. Displaying scepticism or disbelief of CYP's ailments led to caregivers and CYP feeling that CYP's needs had not been met.[31 32 35 39 41 51 60 65]

> I went back there (GP practice) quite a few times and… my GP was trying to convince me that it [Crohn's Disease] was in my head and I was just imagining it. (YP, 24 years,[39])

Two studies (in three reports) of caregivers from deprived areas (one of which also focused on minority ethnic groups) highlighted that parents felt a sense of powerlessness and inferiority in the provider–patient interaction which could prevent them from sharing relevant information or leave them feeling unsupported.[31 32 41]

Continuity of care was considered valuable in building a positive, trusting relationship between YP/caregivers and HCPs[31 53 57 58 60 63 66 73] and was particularly vital for CYP with mental health concerns.[35 51 68 69 74] YP, caregivers and HCP noted that in discussions about sensitive matters, such as mental health, HCP should be careful about the language used and help-seeking should be framed as a healthy and positive behaviour.[44 53 54 65 69 73] Information-giving should be tailored to the individual, for example, YP attending a sexual health service might need more support on their first visit.[55 60] Participants of all types in multiple studies reported that more consultation time

**Table 2** Reported variability/gaps in HCP knowledge

| Variability/gaps in HCP knowledge in treating CYP | Reported by | References | Quality rating of references |
|---|---|---|---|
| **General practitioners** | | | |
| Mental health: presentation of different conditions; enabling CYP to share their concerns; knowledge of available treatment options and CAMHS services; managing potential risks of approaching sensitive topics in front of family members. | CYP HCPs Caregivers | 49 51 68–70 74 | 3 high, 1 medium, 2 low |
| Allergy management and referrals to secondary care. | Caregivers | 66 | 1 medium |
| The needs of primary-aged gender-diverse children and support services available. | Caregivers | 43 | 1 medium |
| Identifying and managing juvenile arthritis. | HCP | 59 | 1 high |
| ADHD aetiology, identification, diagnosis, referral processes, services available. | HCP CYP Caregivers | 42 | 1 medium |
| The experiences and needs of families from ethnic minority groups. | Caregivers CYP | 39 48 | 1 high, 1 medium |
| How to sensitively and effectively address childhood obesity, particularly when caregivers have struggled with their own weight. | Caregivers | 44 | 1 low |
| **Dentistry** | | | |
| Managing children with learning difficulties. | HCP | 96 | 1 low |
| **Health visiting teams** | | | |
| Oral health promotion, culturally specific oral health guidance, knowledge of local dentistry services. | HCP | 36 67 | 2 high |
| Culturally specific advice concerning feeding practices. | Caregivers HCP | 47 | 1 high |
| How to address childhood obesity. | HCPs | 73 | 1 low |

ADHD, attention deficit hyperactivity disorder; CAMHS, Child and Adolescent Mental Health Services; CYP, children and young people; HCP, healthcare professionals.

was needed for sensitive subjects, notably mental health or when support needs were high.[35 41 51 55 60 63 73 74]

### *Confidentiality and family involvement*
YP, particularly those with mental health problems, expressed concern that information about them would be shared with family or other professionals without their consent, as reported in four mixed-quality studies.[35 51 60 74] Parents could be a facilitator or a barrier to mental health consultations with YP: parents could facilitate access by encouraging them to attend and supporting their account; or parents could inhibit the YP from sharing information if the YP did not want to upset them, if they wanted something different from their parent, or their parental relationship was part of the problem.[69 70 74]

### *HCP knowledge and competence*
Studies highlighted multiple areas where HCPs lacked sufficient expertise to manage care (see table 2). GP management of CYPs' mental health was the knowledge and competency gap most often reported by YP, caregivers and HCPs. It included the presentation of different conditions; how to enable CYP to share their concerns; knowledge of available treatment options and CAMHS services; and managing potential risks of approaching

sensitive topics in front of family members (see table 2). If there was a delay or unsuccessful referral in accessing secondary or adult care (see 'Delayed or reject referrals'), then the GP remained the (non-expert) provider of care in the interim.[42 49 68 70 74] Managing physical changes from puberty while waiting for specialist care for gender diversity was a new area where expertise was required.[43]

YP and caregiver trust in HCPs' expertise could diminish when repeated consultations resulted in little improvement or misdiagnosis and were a barrier to seeking further help from primary care, as reported in multiple studies (of predominantly medium/high quality), three focusing on CYP with chronic health conditions and three on ethnic minority groups.[30 31 39 48 59 66] Thus, experiences of communication and trust affected the decision to access care in the future.

### **Barriers affecting equitable access to care**
Specific barriers affecting access to care across themes were mapped for several subpopulations with known higher health needs (see table 3). Multiple trust-related barriers were reported by ethnic minority caregivers and YP resulting from negative past experiences with unfriendly staff, or unsatisfactory support or diagnosis,

**Table 3** Barriers to accessing care for subpopulations of CYP

| Subpopulation (no. of studies) | Reported barriers to access |
|---|---|
| CYP with mental health problems (n=11) | ▶ Decision to access: Stigma related to mental health. CYP believing they would not be taken seriously or would not have a say in their treatment. CYP believing they could self-manage.<br>▶ Reaching and entering services: Caregivers feeling hesitant to persist in asking for support for their child. Unfriendly reception staff. Delayed or rejected referrals to CAMHS or AMH.<br>▶ Communication and trust: A lack of continuity of care and insufficient time in consultations. YP concerns about confidentiality. GPs lacking knowledge in how to manage CYP mental health. |
| CYP from deprived areas (n=8) | ▶ Decision to access: CYP oral health was a lower priority for some caregivers than children's physical health and developmental milestones. Caregivers feeling judged for their parenting or blamed for their child's condition.<br>▶ Reaching and entering services: Caregivers lacking practical resources and non-digital information.<br>▶ Communication and trust: Caregivers feeling a sense of powerless and inferiority in the provider-patient interaction. |
| Looked-after children (n=2) | ▶ Reaching and entering services: Strict non-attendance and de-registration policies.<br>▶ Communication and trust: A lack of continuity of care and insufficient time in consultations. |
| Ethnic minority CYP (n=7) | ▶ Decision to access: A lack of familiarity within the community of the syndrome/illness and stigma related to mental health. Perception of surveillance by healthcare systems. Experiences of stigma and discrimination. Lack of health information in other languages.<br>▶ Reaching and entering services: Unfriendly reception staff. Lack of knowledge of the healthcare (and education) system.<br>▶ Communication and trust: Repeated consultations resulting in little improvement or misdiagnosis. Lack of GP knowledge about the experiences and needs of ethnic minority groups. Health visiting teams lacking knowledge of culturally specific oral health guidance and feeding practices. |
| CYP with SEND (n=5) | ▶ Decision to access: Caregivers feeling judged for their parenting or blamed for their child's condition.<br>▶ Reaching and entering services: Lack of knowledge of the healthcare (and education) system. Delayed or rejected referrals to secondary or adult care. Caregivers feeling hesitant to persist in asking for support for their child.<br>▶ Communication and trust: Dentists lacking knowledge in caring for CYP with learning difficulties. |
| CYP with chronic health problems (n=4) | ▶ Reaching and entering services: Delayed or rejected referrals to secondary care. Caregivers feeling hesitant to persist in asking for support for their child.<br>▶ Communication and trust: Repeated consultations resulting in little improvement or misdiagnosis. Lack of GP knowledge about some childhood chronic health problems. |

CAMHS, Child and Adolescent Mental Health Services; CYP, children and young people; GP, general practitioner; YP, young people.

combined with a need for more accessible and culturally appropriate health information. Many barriers to seeking mental health support were identified by YP, caregivers and HCP, including a lack of patient and HCP awareness of treatment options and organisational processes which diminished relationship-building between YP and HCPs (eg, short appointments, less continuity of care). Common barriers reported across subpopulations were caregivers needing to have the knowledge or confidence to ask for the help they needed or to challenge an HCP whose advice they disagreed with, gaps in HCP knowledge and in communication between primary and secondary care.

## DISCUSSION
### Summary
The review identified high-quality evidence, from multiple studies and informants, that CYP access to primary care was affected by caregivers and YP knowing whether symptoms/conditions could be managed at home or whether healthcare expertise was needed, supporting other studies that show patients must identify themselves as a suitable candidate for healthcare services in order to seek access.[75 76] The NICE review of access also highlighted the importance of CYP having information about the healthcare services available to them.[19] Levels of patients' health and language literacy, access to legitimate health advice via social networks or culturally appropriate resources, and patients' expectations affect equitable and appropriate use of primary care.[11 77 78] This suggests multilingual public health information about childhood symptoms/conditions when and how to seek help should be available online and in public spaces, and professionals who bridge community and primary care services (eg, third sector health workers, health visitors, school nurses, family hub workers) should support caregivers/YP into primary care when they identify healthcare needs and there are known language, cultural or trust-related barriers to accessing services.[19 79–81]

Many high-quality studies suggested that CYP access to services could be improved by making them easier to reach and enter, for example, by extending opening hours and colocated services. Signals that healthcare settings were family-friendly, such as having posters/information designed for CYP in reception, appropriate to the needs of different age groups, and having welcoming and friendly reception staff were quick-wins. Flexibility, for example, having the option to call, drop-in, or use an online system to make an appointment, could facilitate access for caregivers with different needs and preferences in time, communication and support.[82] Wealthier caregivers were able to circumvent blocks to timely secondary care by accessing private healthcare, but this was not possible for all caregivers, suggesting that waiting lists are likely to disadvantage poorer CYP. This is particularly concerning in dentistry where 27000 children were on NHS waiting lists for specialist dental care, assessment or procedures in January 2023.[83] Combined with general practice workforce shortages,[14] increased CYP morbidities[15] and lower caregiver self-efficacy, health and language literacy in deprived areas, the importance of proactive efforts to address inequalities is evident.[82]

Although improving CYP access to mental healthcare is a high policy priority,[8 29] there was strong evidence that YP were reluctant to consult with GPs about mental health concerns without a pre-existing relationship with them. Feelings of fear or embarrassment, experiences of discrimination and/or negative interactions with HCPs, for example, feeling dismissed or unheard, increased CYP's and caregivers' reticence to disclose concerns, a finding mirrored in the NICE review.[19] Prioritising continuity of care for YP to enable trust to develop in a context where GPs are increasingly working part-time and locuming needs consideration.[84–86] Caregivers, YP and HCPs also reported gaps in GPs knowledge/competence in managing CYP mental health, and long-wait times and rejected referrals to secondary care, indicating a need to increase medical training in child and adolescent mental health.[87 88] Although school-based interventions may alleviate concerns for some children, evidence from large-scale mixed-method evaluations suggests that CYP with moderately high emotional needs and those with additional needs (eg, neurodiversity, SEND or difficult family circumstances) may fall through the gaps.[89 90] There are examples of integrated approaches for children with chronic health conditions whereby GPs are supported by specialists which could bridge this gap including in mental health.[91 92] The new role of the primary care CYP mental health practitioner and social prescribing link workers may be able to support CYP waiting for CAMHS, though the evidence for this is not yet known.[93 94]

The review highlighted aspects of primary healthcare experiences that were well evidenced, with multiple studies of high or medium quality across different informants' views. These were experiences of stigma, discrimination and embarrassment as access barriers; access affected by the supply of services; knowledge, confidence and information facilitating CYP's/caregivers' access; and HCPs needing to enable CYP/caregivers to disclose their concerns. However, we also identified several evidence gaps where more research was needed (1) CYP's perspectives on creating family-friendly healthcare settings; (2) CYP's views on the impact of delayed or rejected referrals; (3) high-quality studies on managing confidentiality and parental involvement, including caregivers' perspectives and (4) high-quality studies on experiences of access to oral healthcare and optometry.

## Strengths and limitations

Our review was rigorously conducted and included quality appraisal. Mapping patterns of facilitators/barriers across different subpopulations with higher health needs revealed that access was affected by caregivers' needing to be able to confidently advocate for their child's needs. It also highlighted the multilayered barriers that exist for some groups, including ethnic minority CYP, and the lack of current evidence on access for looked-after children. It extends the findings of the NICE review by highlighting how local healthcare knowledge within communities and social networks affects CYP's and caregivers' decision to seek help, the impact of delays or rejected referrals to secondary care, and areas where HCPs may lack knowledge and competence.

Regarding limitations, we only double-screened 20% of titles/abstracts and we may have missed reports due to the array of terms for primary care, for example, we did not include search terms specific to health visiting, walk-in centres or sexual health clinics. Our definition of access included being able to use healthcare services and have healthcare needs met. Consequently, we viewed communication problems in consultations as part of the negotiation of access and not being referred to secondary care when CYP/caregivers perceived it necessary as a failure to have healthcare needs met. Although including terms specific to patient–doctor communication and referral decision-making would have increased the sensitivity of the review, it would have reduced its specificity and increased the resources needed for screening beyond those that were available. Note, the recommendations on communicating with CYP can be found in the NICE guidelines.[19] We could not screen studies in systematic reviews or search for grey literature due to time and resources constraints, and we may have missed relevant reports, particularly for marginalised groups (eg, lesbian, gay, bisexual, transgender, queer, questioning and ace YP, migrant YP). The impact of workforce barriers to access, for example, recruiting and retaining GPs, were not identified using our search terms and may require specific terms to be added to future reviews.

## Conclusions

The review evidence suggests that four policy priorities to improve equitable CYP access to primary care: (1) encouraging CYP/caregivers into healthcare settings through general practices developing and maintaining links with

community health workers/services, (2) improving CYP/caregivers' understanding of common childhood conditions by providing public health information on common childhood conditions and illnesses in local languages, (3) developing integrated approaches bringing specialist expertise into primary care and (4) addressing paediatric training gaps for medical students, particularly in child and adolescent mental health.

**Acknowledgements** We would like to thank German Alarcon Garavito, Macarena Chepo, Sophie Moniz, Federico Redin and Cecilia Vindrola-Padros from the Rapid Research Evaluation and Appraisal (RREAL) Lab for their contributions to the screening for this review.

**Contributors** LH, CP, JW, KH and SM contributed to the study's conception, LH conducted the searches, and LH and EA completed the screening. LH, EA and CP conducted the data extraction and quality appraisal, and LH and EA carried out the data synthesis. LH led and EA contributed to drafting the manuscripts and all authors provided critical revisions and editing. All authors reviewed the manuscript. LH is responsible for the overall content as guarantor.

**Funding** This study was funded by the National Institute for Health and Care Research (NIHR) through the Children and Families Policy Research Unit (PR-PRU-1217-21301).

**Disclaimer** The views expressed are those of the authors and not necessarily those of the NIHR or the Department of Health and Social Care.

**Competing interests** None declared.

**Patient and public involvement** Patients and/or the public were not involved in the design, or conduct, or reporting, or dissemination plans of this research.

**Patient consent for publication** Not applicable.

**Provenance and peer review** Not commissioned; externally peer reviewed.

**Data availability statement** All data relevant to the study are included in the article or uploaded as online supplemental information.

**ORCID iDs**
Lauren Herlitz http://orcid.org/0000-0003-2497-9041
Claire Powell http://orcid.org/0000-0002-6581-0165
Kevin Herbert http://orcid.org/0009-0008-4354-7811
Stephen Morris http://orcid.org/0000-0002-5828-3563
Jenny Woodman http://orcid.org/0000-0002-9403-4177

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
