## [Reviewer comments · BMJ Open]

ARTICLE DETAILS

TITLE (PROVISIONAL)	Access to primary care for children and young people (CYP) in the UK: a scoping review of CYP's, caregivers', and healthcare professionals' views and experiences of facilitators and barriers
AUTHORS	Herlitz, Lauren; Ashford, Emily; Powell, Claire; Herbert, Kevin; Morris, Stephen; Woodman, Jenny

VERSION 1 – REVIEW

REVIEWER	Richard Churchill Nottingham City General Practice Alliance
REVIEW RETURNED	15-Jan-2024

GENERAL COMMENTS	The paper is generally well written and presented. Methodology is appropriate and clearly described. Themes described are consistent with other studies and findings are discussed appropriately. The following points need to be addressed: Major Points The scope of the review goes beyond the stated objective of describing facilitators and barriers to access to UK primary care, in that it also covers communication issues in the consultation and referrals to secondary care. Whilst these are important issues, and there are some inter-relationships between them (for example, previous experience influencing future access decisions as alluded to on p18 line 34) it is unclear whether the search strategy focussed sufficiently on these additional factors. The age range included (children and young people up to the age of 25) are not a homogeneous group in terms of accessing services: younger children access services based on parents decisions, and parents also negotiate access; teenagers start to become independent help-seekers and also need to learn to navigate health systems independently; young adults are likely to be more confident in this regard. Another example is that 'child-friendly' waiting areas may not be as inviting for teenagers or young people. This heterogeneity needs to be acknowledged explicitly in the introduction or discussion – although I noted that some of the results were specifically described in relation to these separate sub-groups. There was a surprising failure to reference or acknowledge the NICE Guideline on Babies, Children and Young People's Experience of Healthcare (NG204) published in 2021 which included a systematic review of Accessing Healthcare (https://www.nice.org.uk/guidance/ng204/evidence/o-accessing-healthcare-pdf-331430552538) Although there are differences in scope (the NICE review was not limited to primary care, had an upper age limit of 18; focussed on
--

	experiences of CYP themselves; and had a different search time period) I would have expected the current report to include most of the studies included in the NICE review. I noted the following apparent omissions whilst accepting that there may have been reasons why these had been excluded: Ali, Nasreen, McLachlan, Niel, Kanwar, Shama, Randhawa, Gurch, Pakistani young people's views on barriers to accessing mental health services, International Journal of Culture and Mental Health, 10, 33- 43, 2017 Best, Gil-Rodriguez, Manktelow and Taylor, Seeking Help From Everyone and No- One: Conceptualizing the Online Help- Seeking Process Among Adolescent Males, Qualitative Health Research, 26, 1067-1077, 2016 Fargas-Malet, Montserrat, McSherry, Dominic, The mental health and help- seeking behavior of children and young people in care in Northern Ireland: Making services accessible and engaging, British Journal of Social Work, 48, 578-595, 2018 Haig-Ferguson, A., Loades, M., Whittle, C., Read, R., Higson-Sweeney, N., Beasant, L., Starbuck, J., Crawley, E., "It's not one size fits all"; the use of videoconferencing for delivering therapy in a Specialist Paediatric Chronic Fatigue Service, Internet Interventions, 15, 43- 51, 2019 Heath, G., Greenfield, S., Redwood, S., The meaning of 'place' in families' lived experiences of paediatric outpatient care in different settings: A descriptive phenomenological study, Health and Place, 31, 46-53, 2015 Robards, F., Kang, M., Usherwood, T., Sanci, L., How Marginalized Young People Access, Engage With, and Navigate Health-Care Systems in the Digital Age: Systematic Review, Journal of Adolescent Health, 365-381, 2018 Sime, D., 'I think that Polish doctors are better': Newly arrived migrant children and their parents' experiences and views of health services in Scotland, Health and Place, 30, 86-93, 2014 Whittle, N., Macdonald, W., Bailey, S., A Study of Young Offenders' Perceptions of Health and Health Care Services in Custody and in the Community, Journal of Correctional Health Care, 2, 2, 2012 Even if there is a clear rationale for exclusion of the above studies, the current paper should discuss similarities and differences between the findings of the reviews. Minor Points There is little reference to the role of the new range of healthcare professionals working in primary care. Whilst these will not feature significantly in previous research studies they are likely to influence future access for CYP (although many are currently only trained in adult health issues). Although the authors state that workforce supply barriers affect all patients equally (p22 line 16) I would suggest that they may differentially disadvantage some groups more than others. (see British Journal of General Practice 2022; 72 (716): 123. DOI: https://doi.org/10.3399/bjgp22X718685) Discussion of continuity issues could reference the recent paper linked to the NICE Guideline: Archives of Disease in Childhood 2023-10 Journal article DOI: 10.1136/archdischild-2022-324456
--	--

REVIEWER	Ben Hughes University of Bolton, School of Health and Society
REVIEW RETURNED	19-Jan-2024

GENERAL COMMENTS	Thank you for your manuscript and the opportunity to read such a strong paper on such an important topic. I think this is a very valuable study and you've presented your search strategy and review of the literature nicely. The study is very clearly outlined at each stage and you've provided a good appraisal of the literature with a lot of useful supplementary evidence. There are key learning points, which are really important to highlight in order to improve access to primary care and your findings are important for a range of CYP from different ages and backgrounds. The paper reads very well and I hope this will be a good foundation for further research in this area. I've provided a few points for your consideration and I'm happy for you to provide a rationale for not amending/including them if you feel they are not relevant:  1. Page 5 – Introduction – Would it be worth stating the definition of CYP in the introduction? There isn't a definitive definition in the UK and so many organisations define their own upper age, which has been rising over recent years. This would help justify your upper age of 25 years for your study. 2. Page 6 – Document selection – Would it be worth indicating which authors were involved in different stages of document selection so this process is clear, especially as you've included an author's initials in the Quality Appraisal section? 3. Page 23 – Discussion – I think you've provided a very good discussion and summary of the literature and I think it would be useful to provide some critique of it. You've clearly highlighted where the quality of the sources are low, medium, or high and a bit more discussion of this, and the impact of it, will be useful in your discussion and appraisal of the literature.
---

VERSION 1 – AUTHOR RESPONSE

Reviewer: 1

Dr. Richard Churchill, Nottingham City General Practice Alliance

Comments to the Author:

The paper is generally well written and presented. Methodology is appropriate and clearly described. Themes described are consistent with other studies and findings are discussed appropriately.

The following points need to be addressed:

Major Points

The scope of the review goes beyond the stated objective of describing facilitators and barriers to access to UK primary care, in that it also covers communication issues in the consultation and referrals to secondary care. Whilst these are important issues, and there are some inter-relationships between them (for example, previous experience influencing future access decisions as alluded to on p18 line 34) it is unclear whether the search strategy focussed sufficiently on these additional factors.

We have taken a broad definition of 'access' in our review, including being able to use healthcare services and have healthcare needs met. Consequently, we viewed communication problems in the consultation as part of the negotiation of access, and not being referred to secondary care when it was perceived as needed a failure to have healthcare needs met. Although including terms specific to

patient-doctor communication and referral decision-making would have increased the sensitivity of the review, it would have reduced its specificity and increased the resources needed for screening beyond those that were available. We have noted these points in the discussion under limitations.

The age range included (children and young people up to the age of 25) are not a homogeneous group in terms of accessing services: younger children access services based on parents decisions, and parents also negotiate access; teenagers start to become independent help-seekers and also need to learn to navigate health systems independently; young adults are likely to be more confident in this regard. Another example is that 'child-friendly' waiting areas may not be as inviting for teenagers or young people. This heterogeneity needs to be acknowledged explicitly in the introduction or discussion – although I noted that some of the results were specifically described in relation to these separate sub-groups.

We have added acknowledge of heterogeneity due to age and development to the introduction. We have added “CYP age focus” to the study characteristics section and we have included additional information on participants’ age in the findings (see tracked changes). We have also added a comment in the discussion.

There was a surprising failure to reference or acknowledge the NICE Guideline on Babies, Children and Young People’s Experience of Healthcare (NG204) published in 2021 which included a systematic review of Accessing Healthcare (<https://www.nice.org.uk/guidance/ng204/evidence/o-accessing-healthcare-pdf-331430552538>)

Although there are differences in scope (the NICE review was not limited to primary care, had an upper age limit of 18; focussed on experiences of CYP themselves; and had a different search time period) I would have expected the current report to include most of the studies included in the NICE review. I noted the following apparent omissions whilst accepting that there may have been reasons why these had been excluded:

Ali, Nasreen, McLachlan, Niel, Kanwar, Shama, Randhawa, Gurch, Pakistani young people's views on barriers to accessing mental health services, International Journal of Culture and Mental Health, 10, 33- 43, 2017

Best, Gil-Rodriguez, Manktelow and Taylor, Seeking Help From Everyone and No- One: Conceptualizing the Online Help- Seeking Process Among Adolescent Males, Qualitative Health Research, 26, 1067-1077, 2016

Fargas-Malet, Montserrat, McSherry, Dominic, The mental health and help- seeking behavior of children and young people in care in Northern Ireland: Making services accessible and engaging, British Journal of Social Work, 48, 578-595, 2018

Haig-Ferguson, A., Loades, M., Whittle, C., Read, R., Higson- Sweeney, N., Beasant, L., Starbuck, J., Crawley, E., "It's not one size fits all"; the use of videoconferencing for delivering therapy in a Specialist Paediatric Chronic Fatigue Service, Internet Interventions, 15, 43- 51, 2019

Heath, G., Greenfield, S., Redwood, S., The meaning of 'place' in families' lived experiences of paediatric outpatient care in different settings: A descriptive phenomenological study, Health and Place, 31, 46-53, 2015

Robards, F., Kang, M., Usherwood, T., Sanci, L., How Marginalized Young People Access, Engage With, and Navigate Health-Care Systems in the Digital Age: Systematic Review, Journal of Adolescent Health, 365-381, 2018

Sime, D., 'I think that Polish doctors are better': Newly arrived migrant children and their parents' experiences and views of health services in Scotland, Health and Place, 30, 86-93, 2014

Whittle, N., Macdonald, W., Bailey, S., A Study of Young Offenders' Perceptions of Health and Health Care Services in Custody and in the Community, Journal of Correctional Health Care, 2, 2, 2012

Even if there is a clear rationale for exclusion of the above studies, the current paper should discuss similarities and differences between the findings of the reviews.

Thank you so much for directing us towards this review. We have included the review in our introduction and highlighted similarities to and extensions to the scope of the NICE review (the reviews do not differ in their findings). For your interest, the table below presents the studies from the NICE review, and the application of our review criteria. There is one study on newly migrant children that was not picked up by our search strategy that would have been relevant, and we have noted migrants as a group that the review strategy might have missed in the limitations section. We had noted the exclusion of systematic reviews in the limitations section of the discussion, but we have now added it also to the inclusion/exclusion criteria in the methods for clarity.

Included study from NICE guidelines	Decision taken in our review
Ali et al (2017)	This study was screened at full-text. It was excluded on healthcare setting as it focused on secondary care.
Best et al (2016)	This was not picked up in our search. It is likely because the T/A does not pick up terms for primary care.
Dickson (2015)	Included in our review.
Diwakar et al (2019)	Included in our review.
Fargas-Malet et al (2018)	This study was not picked up in our search. It is likely because the T/A does not pick up terms for primary care.
Haig-Ferguson et al (2019)	This study was screened at full-text. It was excluded on healthcare setting as it focused on secondary care.
Heath et al (2015)	This study was screened at full-text. It was excluded on healthcare setting as it focused on secondary care.
Leavey et al (2011)	This study was not picked up in our search as it was published before 2012.
Neill et al (2016)	Included in our review.
Robards et al (2018)	This study was screened on T/A. It was excluded on study design (systematic review) and healthcare setting (focused on any healthcare setting).
Sime (2014)	This study was not picked up in our search. It is likely because the T/A does not pick up terms for primary care. On reading the full-text, this would have been relevant for our study.
Turnball et al (2010)	This study was not picked up in our search because it is published before 2012.
Walsh et al (2011)	This study was not picked up in our search because it is published before 2012.
Whittle et al (2012)	This study was not picked up in our search. It is likely because the T/A does not pick up terms for primary care.

Minor Points

There is little reference to the role of the new range of healthcare professionals working in primary

care. Whilst these will not feature significantly in previous research studies they are likely to influence future access for CYP (although many are currently only trained in adult health issues).

Thanks for this helpful comment. We have made reference to the expanded workforce in the discussion.

Although the authors state that workforce supply barriers affect all patients equally (p22 line 16) I would suggest that they may differentially disadvantage some groups more than others. (see British Journal of General Practice 2022; 72 (716): 123. DOI: <https://doi.org/10.3399/bjgp22X718685>)

We did not intend to imply that workforce supply barriers affect all patients equally; indeed, we have argued the opposite point earlier in the discussion. To avoid confusion, we have revised the text related to this point.

Discussion of continuity issues could reference the recent paper linked to the NICE Guideline: Archives of Disease in Childhood 2023-10 | Journal article DOI: 10.1136/archdischild-2022-324456

Thank you, this reference is highly relevant and we have added it to the discussion.

Reviewer: 2

Dr. Ben Hughes, University of Bolton

Comments to the Author:

Thank you for your manuscript and the opportunity to read such a strong paper on such an important topic. I think this is a very valuable study and you've presented your search strategy and review of the literature nicely. The study is very clearly outlined at each stage and you've provided a good appraisal of the literature with a lot of useful supplementary evidence. There are key learning points, which are really important to highlight in order to improve access to primary care and your findings are important for a range of CYP from different ages and backgrounds. The paper reads very well and I hope this will be a good foundation for further research in this area. I've provided a few points for your consideration and I'm happy for you to provide a rationale for not amending/including them if you feel they are not relevant:

1. Page 5 – Introduction – Would it be worth stating the definition of CYP in the introduction? There isn't a definitive definition in the UK and so many organisations define their own upper age, which has been rising over recent years. This would help justify your upper age of 25 years for your study.

We have added the age range in the first para – 0 – 25 years is the range defined in the NHS Long Term Plan.

2. Page 6 – Document selection – Would it be worth indicating which authors were involved in different stages of document selection so this process is clear, especially as you've included an author's initials in the Quality Appraisal section?

Some of the reviewers who were involved in T/A screening are not authors and instead are acknowledged in the contributions. For consistency, we have removed the initials from the quality appraisal section. Readers can now refer to the contributions section for full information.

3. Page 23 – Discussion – I think you've provided a very good discussion and summary of the literature and I think it would be useful to provide some critique of it. You've clearly highlighted where the quality of the sources are low, medium, or high and a bit more discussion of this, and the impact of it, will be useful in your discussion and appraisal of the literature.

Thanks for this constructive feedback. We have added an additional paragraph to the Discussion to address this point.

VERSION 2 – REVIEW

REVIEWER	Richard Churchill Nottingham City General Practice Alliance
REVIEW RETURNED	11-Mar-2024

GENERAL COMMENTS	The changes that have been made in this revised version of the paper have fully addressed the comments made by reviewers on the original version. It reads well and adequately discusses strengths and limitations. It should be of interest to a wide range of readers.
--

REVIEWER	Ben Hughes University of Bolton, School of Health and Society
REVIEW RETURNED	08-Mar-2024

GENERAL COMMENTS	Thank you for your revised manuscript. I can see that you've taken on board the comments from both reviewers and I feel that this has added to the quality of your article. You have clarified your definition of CYP and added some useful references to support your later discussion. I feel that your additional sections, and particularly the references to the NICE review, have helped strengthen your manuscript further and thank you for the responses to my particular comments around the role of different contributors and also the addition of the paragraph in your Discussion section. What I felt was already a strong paper has, in my view, been lifted further by your amendments. I think your structure and writing style are appropriate and accessible and your methodology and results help you to draw important conclusions. Thank you for the time and effort you have put into this paper and what you have added to this important topic.
---